# An ultralong CDRH2 in HCV neutralizing antibody demonstrates structural plasticity of antibodies against E2 glycoprotein

Andrew I Flyak[1], Stormy E Ruiz[1,2], Jordan Salas[2], Semi Rho[1], Justin R Bailey[2], Pamela J Bjorkman[1]*

[1]Division of Biology and Biological Engineering, California Institute of Technology, Pasadena, United States; [2]Department of Medicine, Johns Hopkins University School of Medicine, Baltimore, United States

**Abstract** A vaccine protective against diverse HCV variants is needed to control the HCV epidemic. Structures of E2 complexes with front layer-specific broadly neutralizing antibodies (bNAbs) isolated from HCV-infected individuals, revealed a disulfide bond-containing CDRH3 that adopts straight (individuals who clear infection) or bent (individuals with chronic infection) conformation. To investigate whether a straight versus bent disulfide bond-containing CDRH3 is specific to particular HCV-infected individuals, we solved a crystal structure of the HCV E2 ectodomain in complex with AR3X, a bNAb with an unusually long CDRH2 that was isolated from the chronically-infected individual from whom the bent CDRH3 bNAbs were derived. The structure revealed that AR3X utilizes both its ultralong CDRH2 and a disulfide motif-containing straight CDRH3 to recognize the E2 front layer. These results demonstrate that both the straight and bent CDRH3 classes of HCV bNAb can be elicited in a single individual, revealing a structural plasticity of *VH1-69*-derived bNAbs.

*For correspondence: bjorkman@caltech.edu

## Introduction

HCV infections are on the rise in the United States, reflecting increasing rates of opioid addiction (*Zibbell et al., 2018*). An HCV vaccine is urgently needed to control the epidemic, but vaccine development is challenging due to the enormous genetic diversity of the HCV envelope proteins (*Yusim et al., 2010*). The HCV genome encodes two structural proteins, E1 and E2, that associate to form a noncovalent heterodimer, E1E2 (*Freedman et al., 2016*). Potent bNAbs isolated from HCV-infected individuals predominantly target conserved epitopes in the front layer of the E2 glycoprotein. The majority of bNAbs that bind to the front layer are derived from *VH1-69* genes (*Tzarum et al., 2019*), which are also associated with bNAbs that target conserved epitopes on influenza virus and HIV-1 envelope glycoproteins (*Chen et al., 2019*).

We recently described crystal structures of two *VH1-69* bNAbs, HEPC3 and HEPC74, isolated from individuals who spontaneously cleared HCV infection (*Flyak et al., 2018*). Both bNAbs utilized a disulfide motif in their CDRH3 regions to recognize a conserved epitope in the front layer of E2. While the HEPC3 and HEPC74 CDRH3 loops adopted a straight ß-hairpin conformation, the *VH1-69*-encoded AR3A and AR3C bNAbs that were isolated from an individual with a chronic HCV infection included bent CDRH3 loops that contained an analogous disulfide motif (*Kong et al., 2013*). Since the two bNAbs with straight CDRH3s were isolated from individuals who spontaneously cleared HCV infection and the two bNAbs with bent CDRH3s were isolated from a single chronically-infected individual, we wondered if some individuals are naturally predisposed to make

antibodies with straight or bent CDRH3s and/or whether the straight CDRH3 conformation was related to the ability to clear HCV infection. Among bNAbs isolated from a chronically-infected individual (*Law et al., 2008*), we found AR3X, a *VH1-69*-encoded antibody that included a CDRH3 with a disulfide motif and an unusually long 14-amino acid-long insertion in CDRH2 (*Figure 1A*). AR3X provided an opportunity to explore the structural plasticity of *VH1-69*-derived anti-HCV bNAbs with a disulfide-containing CDRH3 and to determine the impact of a long CDRH2 insertion on the recognition of the conserved epitope in E2 front layer.

## Results

The most likely scenario resulting in the insertion into the CDRH2 of AR3X involves a duplication event, as the CDRH2 insertion has 69% identity with the N-terminal sequence preceding the CDRH2 (*Figure 1B*). Similar to other front layer-specific bNAbs with the CDRH3 disulfide motif (*Figure 1E*), the cysteines in the AR3X CDRH3 region are encoded by the human D gene segment 15 (IGHD2-15) (*Figure 1C*). The C-terminal portion of the AR3X CDRH3 is likely encoded by human J-gene segment 3*02 (J3*02). Not including the 14-amino acid insertion in CDRH2, AR3X shares 91% nucleotide

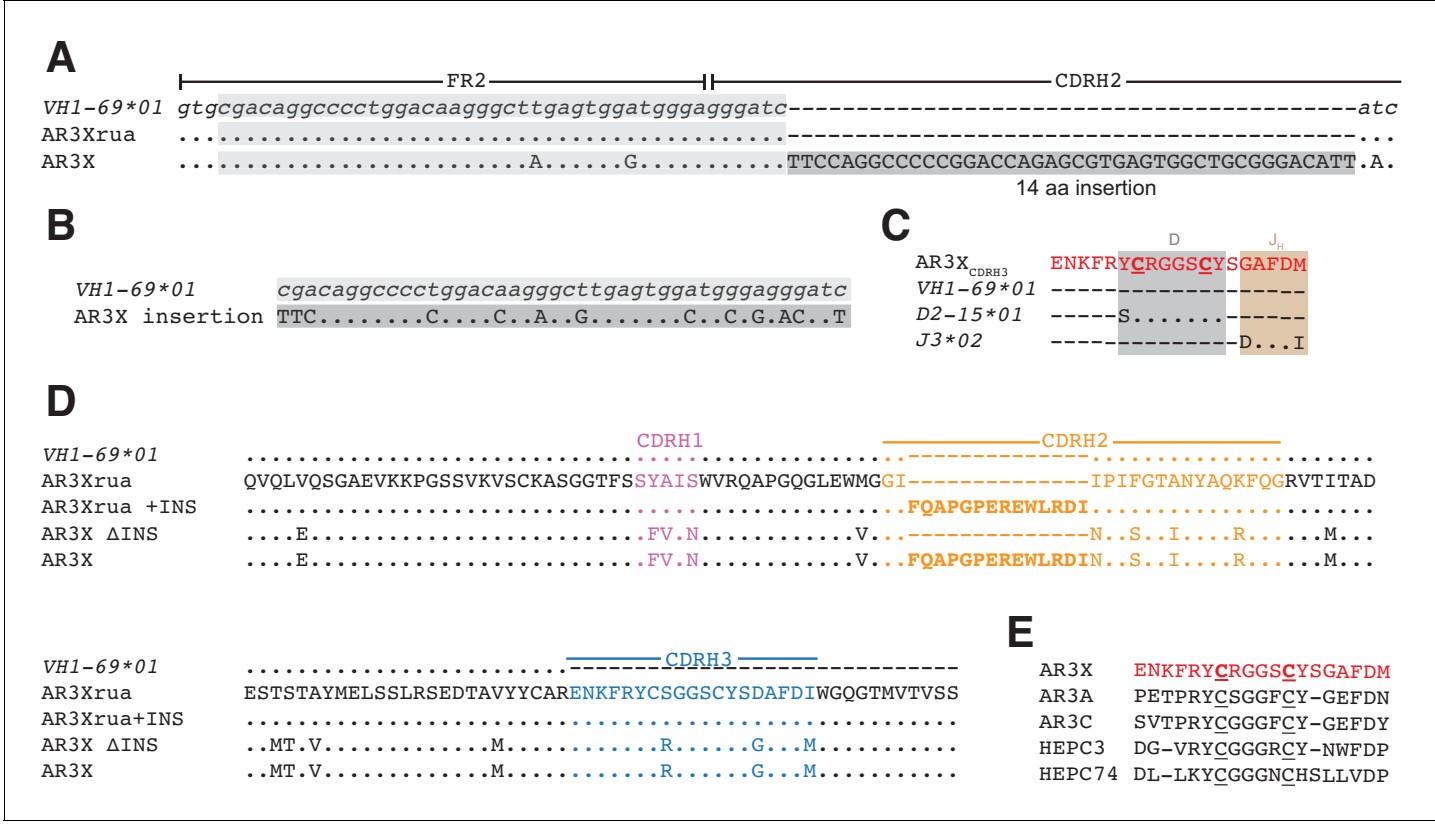

**Figure 1.** AR3X includes a 14-residue insertion in CDRH2. (a) Sequence alignment of a portion of the heavy chain variable region gene sequences of AR3X and the AR3X germline precursor (AR3Xrua) (uppercase letters) and the *VH1-69* gene segment (lowercase letters). The CDRH2 insertion is indicated by a dark gray box with the position of the potential duplication site indicated by a light gray box. CDR loops were defined based on Kabat nomenclature *Kabat and National Institutes of Health (U.S.). Office of the Director, 1991*). Dots indicate identical nucleotides and dashes indicate gaps. (b) Sequence alignment of the CDRH2 insertion and the potential duplication origin site in *VH1-69*. (c) Amino acid sequence alignment of the AR3X CDRH3 and the AR3X germline precursor genes determined by IMGT/V-QUEST. Dots indicate identical amino acids and dashes indicate regions encoded by other gene segments or N-nucleotide additions. Two cysteines encoded by the D gene segment are highlighted in bold and underscored. (d) Amino acid sequence alignment of the heavy chain variable region sequences of AR3X, AR3X ΔINS (AR3X without insertion), AR3Xrua (germline precursor of AR3X), and AR3Xrua + INS (germline precursor of AR3X with insertion). CDR loops were defined based on Kabat nomenclature and colored purple (CDRH1), orange (CDRH2), and blue (CDRH3), with the CDRH2 insertion highlighted in bold. Dots indicate identical amino acids and dashes indicate gaps. (e) Alignment of AR3X, AR3A, AR3C, HEPC3, and HEPC74 CDRH3 sequences. The AR3X sequence is highlighted in red and the two cysteines in each CDRH3 are underscored.

identity with the $V_H1$-$69$ gene segment and includes 17 somatic mutations (*Figure 1D*). To investigate the importance of the CDRH2 insertion and the effects of somatic mutations on AR3X binding and neutralization, we generated a panel of AR3X variants: AR3X ΔINS (AR3X without the CDRH2 insertion), AR3Xrua (germline precursor of AR3X, which lacks the CDRH2 insertion and somatic mutations), and AR3Xrua + INS (germline precursor of AR3X with the CDRH2 insertion) (*Figure 1D*).

We evaluated the binding of AR3X and AR3X variants to a panel of E2 ectodomain (E2ecto) proteins representing the E2 envelopes from 19 HCV genotype 1 strains. We also tested the binding of AR3X and AR3X variants to E2ecto proteins from genotypes 2, 3, 4, 5, and 6 strains. AR3X recognized all 19 E2 envelopes from genotype 1 including the 1a116 strain, which was not recognized by other front layer-specific bNAbs that include the CDRH3 disulfide motif (*Figure 2A*, *Figure 2—figure supplement 1*; *Flyak et al., 2018*). AR3X also recognized E2 envelopes from genotypes 2, 3, 4, 5, and 6 (*Figure 2A*). In contrast to mature AR3X, the AR3X ΔINS protein that lacks the CDRH2 insertion bound only 4 of the 25 variants, indicating that the CDRH2 insertion mediates the breath of

**A** Binding (EC$_{50}$; ng/mL)

| HCV genotype | Strain | AR3X | AR3X ΔINS | AR3Xrua+INS | AR3Xrua |
|---|---|---|---|---|---|
| 1 | 1a09 | 79 | 1,528 | > | > |
| | 1a31 | 155 | > | > | > |
| | 1a38 | 42 | > | > | > |
| | 1a53 | 20 | > | > | > |
| | 1a72 | 44 | > | > | > |
| | 1a80 | 55 | 1,495 | > | > |
| | 1a116 | 1,370 | > | > | > |
| | 1a123 | 250 | > | > | > |
| | 1a129 | 100 | > | > | > |
| | 1a142 | 145 | 4,479 | > | > |
| | 1a154 | 114 | > | > | > |
| | 1a157 | 14 | 24 | 876 | > |
| | 1b09 | 96 | > | > | > |
| | 1b14 | 105 | > | > | > |
| | 1b21 | 455 | > | > | > |
| | 1b34 | 117 | > | > | > |
| | 1b38 | 130 | > | > | > |
| | 1b52 | 74 | > | > | > |
| | 1b58 | 107 | > | > | > |
| 2 | JFH1 | 80 | > | > | > |
| | J6 | 163 | > | > | > |
| 3 | S52 | 73 | > | > | > |
| 4 | ED43 | 79 | > | > | > |
| 5 | SA13 | 135 | > | > | > |
| 6 | HK6a | 60 | > | > | > |

Legend (A): 10 - 100 ng/mL | 1,000 - 10,000 ng/mL | 100 - 1,000 ng/mL | > >10,000 ng/mL

**B** Neutralization (IC$_{50}$; µg/mL)

| Strain | AR3X | AR3X ΔINS | AR3Xrua+INS | AR3Xrua |
|---|---|---|---|---|
| 1a09 | 2.2 | > | > | > |
| 1a31 | 1.4 | > | > | > |
| 1a38 | 16.1 | > | > | > |
| 1a53 | 5.4 | > | > | > |
| 1a72 | 10.3 | > | > | > |
| 1a80 | 51.1 | > | > | > |
| 1a116 | > | > | > | > |
| 1a123 | 3.2 | > | > | > |
| 1a129 | > | > | > | > |
| 1a142 | 16.2 | > | > | > |
| 1a154 | 4.6 | > | > | > |
| 1a157 | 15.5 | > | > | > |
| 1b09 | 1.7 | > | > | > |
| 1b14 | 9.5 | > | > | > |
| 1b21 | 1.2 | > | > | > |
| 1b34 | 17.6 | > | > | > |
| 1b38 | 6.9 | > | > | > |
| 1b52 | 11.3 | > | > | > |
| 1b58 | 27.6 | > | > | > |

Legend (B): 1.0 - 10.0 µg/mL | 10.0 - 100.0 µg/mL | > >100.0 µg/mL

**Figure 2.** The CDRH2 insertion in AR3X is required for maximal binding and broad neutralization. (a) Heat map showing the binding of AR3X and its variants to a panel of HCV E2ecto proteins. The EC$_{50}$ value for each E2ecto-mAb combination is shown, with dark red, orange, yellow, or white shading indicating high, intermediate, low, or no detectable binding, respectively. The > symbol indicates EC$_{50}$s greater than 10 µg/mL or EC$_{50}$s in which the OD$_{450}$ values at the highest antibody concentration tested were lower than 0.5. One experiment representative of two independent experiments is shown. (b) Heat map showing neutralization activities of AR3X and AR3X variants measured using a panel of genotype 1 HCVpp. IC$_{50}$ values for each virus-mAb combination are shown. The > symbol indicates IC$_{50}$s greater than 100 µg/mL or IC$_{50}$s in which the percent neutralization at the highest antibody concentration tested was lower than 50%.

The online version of this article includes the following figure supplement(s) for figure 2:

**Figure supplement 1.** Binding of AR3X and its variants to a panel E2ecto proteins.
**Figure supplement 2.** Neutralization activities of AR3X and its variants against a panel of genotype 1 HCVpp.

binding. While AR3Xrua failed to bind any E2ecto proteins, AR3Xrua + INS recognized 1 of the 25 variants, further highlighting the importance of the CDRH2 insertion in initial recognition of the E2 antigen by naïve B cells. The fact that AR3Xrua + INS only bound to one HCV strain, whereas mature AR3X recognized all strains, indicated that somatic mutations, in addition to the CDRH2 insertion, are required for breath of binding and optimal E2 recognition. Consistent with our previous studies in which the strain 1a157 E2ecto envelope was recognized by HEPC3, HEPC74, AR3C and their germline precursors (*Flyak et al., 2018*), AR3X and two AR3X variants (AR3X ΔINS, AR3Xrua + INS) also bound to 1a157, suggesting that immunogens based on the genotype 1 1a157 ectodomain sequence could be used to stimulate the development of potent front layer-specific bNAbs (*Figure 2A*, *Figure 2—figure supplement 1*).

To evaluate the neutralization breadth of AR3X variants, we evaluated antibodies in an in vitro neutralization assay using a panel of 19 genotype 1 HCV pseudoparticles (HCVpp) that represents 94% of the amino acid polymorphisms present at >5% frequency in a reference panel of 643 genotype 1 HCV isolates from GenBank (*Munshaw et al., 2012*). Only mature AR3X exhibited neutralization activity, neutralizing 17 of 19 HCV strains (*Figure 2B*, *Figure 2—figure supplement 2*). The neutralization breadth of AR3X (89%) was slightly lower than the breath of AR3C bNAb (100%) (*Flyak et al., 2018*), which was isolated from the same HCV-infected individual (*Law et al., 2008*). AR3X variants failed to neutralize HCV isolates, suggesting that both the CDRH2 insertion and somatic mutations are required for the broad neutralization activity of AR3X.

We and others described two classes of *VH1-69* bNAbs with a CDRH3 disulfide motif: bNAbs with a straight CDRH3 (HEPC3 and HEPC74) and bNAbs with a kinked CDRH3 (AR3A and AR3C) (*Flyak et al., 2018*; *Kong et al., 2013*; *Tzarum et al., 2019*; *Figure 3*). To determine to which class AR3X belongs, we determined the crystal structure of AR3X in complex with E2ecto from the 1b09 HCV strain (*Figure 4*, *Figure 4—figure supplement 1*). The 2.2 Å AR3X-E2ecto structure demonstrated that, similar to previously-characterized HCV bNAbs that recognize the neutralizing face of E2 (*Flyak et al., 2018*; *Kong et al., 2013*; *Tzarum et al., 2019*), AR3X binds to the conserved epitope in the E2 front layer (*Figure 4A*). The AR3X CDRH3 loop contains two cysteines that form a disulfide bond, as seen in multiple other E2 front layer-binding bNAbs, and the AR3X CDRH3 adopts the straight conformation we previously described in the HEPC3 and HEPC74 bNAbs that were

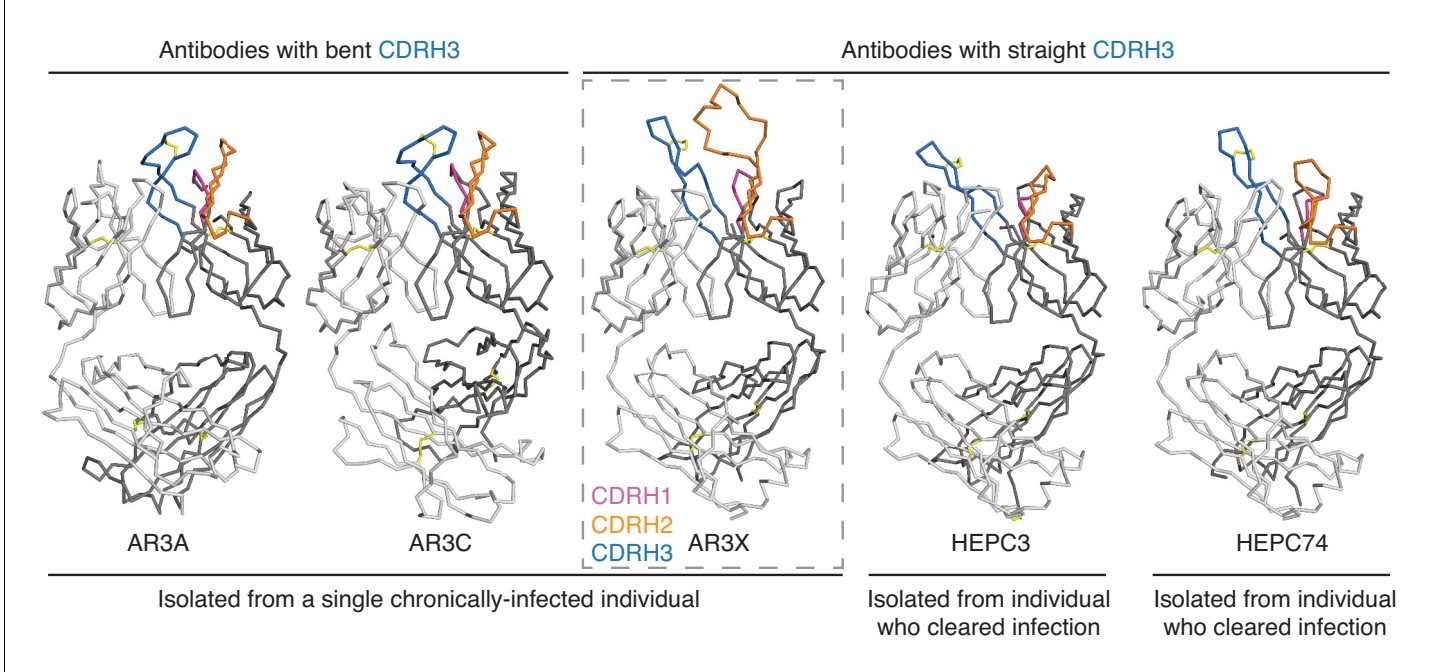

**Figure 3.** The shared CDRH3 motif in E2 front layer-specific HCV bNAbs adopts different orientations. Fab structures in liganded state of AR3A (PDB 6BKB), AR3C (PDB 4MWF), AR3X (this paper), HEPC3 (PDB 6MEI), and HEPC74 (PDB 6MEH). The structures were superimposed on their V$_H$ domains. Protein backbones are shown as ribbons and CDR loops are purple (CDRH1), orange (CDRH2), and blue (CDRH3).

isolated from an individual who cleared HCV infection (*Flyak et al., 2018*; *Figure 3*). By contrast, the CDRH3s of AR3A and AR3C, which were isolated from the same HCV-infected individual as AR3X (*Law et al., 2008*), are bent (*Kong et al., 2013*; *Tzarum et al., 2019*). The tip of the AR3X CDRH3 loop interacts with the same conserved residues in the front layer of E2 as the CDRH3 tips in the other HCV bNAbs (*Figure 4e*, *Figure 4—figure supplement 2*).

Overall, AR3X has a similar binding footprint to the footprints of HEPC3, HEPC74, AR3C, and AR3A, sharing multiple contact residues in the front layer and CD81 receptor-binding loop (*Figure 4—figure supplement 2*). As also found for these other front layer-specific bNAbs, AR3X's contacts with E2ecto almost exclusively involved $V_H$ domain residues, burying 1,250 $\text{Å}^2$ (98% of the total Fab buried surface area; BSA) (*Figure 4B*), with the CDRH3 accounting for 44.5% (556 $\text{Å}^2$) of the total BSA on the $V_H$ domain (*Figure 4B*, *Figure 4—figure supplement 2*). However, in contrast to other front layer-specific bNAbs in which the CDRH3 plays a dominant role in the interactions with E2 envelope (*Flyak et al., 2018*; *Kong et al., 2013*; *Tzarum et al., 2019*), the main contributor to the AR3X-E2ecto binding interface was CDRH2, which accounted for 48.2% (602 $\text{Å}^2$) of the total BSA of the $V_H$ domain, with the majority of the binding footprint provided by the CDRH2 insertion (45.4% or 567 $\text{Å}^2$ of total BSA of the $V_H$ domain) (*Figure 4C*).

We next investigated the frequency of antibodies with 14-residue CDRH2 insertions. While the size of an insertion or deletion in human antibody genes ranges from 3 to 33 nucleotides (*Kanyavuz et al., 2019*), AR3X has a unusually long 42-nucleotide insertion, which results in a 31-residue CDRH2 (Kabat definition: [*Kabat and National Institutes of Health (U.S.). Office of the Director, 1991*]). According to the abYsis database (*Swindells et al., 2017*), a typical human CDRH2 is 17 residues (relative frequency 67%) (*Figure 4D*), and CDRH2s longer than 20 residues are rare (relative frequency <1%). To our knowledge, AR3X with its 31-residue CDRH2 represents the longest CDRH2 among antibody structures available in the Protein Data Bank (PDB).

Although the CDH3s of AR3X, AR3A, AR3C, HEPC3, and HEPC74 CDRH3s make similar binding footprints on the E2 surface (*Figure 5*), the difference in Fab approach angles and the presence of the long insertion in the AR3X CDRH2 result in different footprints on E2 for the $V_H1$-69–encoded CDRH2 loops of the bNAbs: AR3X CDRH2 contacts the C-terminus of the E2 α1-helix, the portion of the E2 front layer between the α-helix, variable region 2 (residues 446–448), and the back layer of E2 (residues 444, 445) (*Figure 4F* and *Figure 4—figure supplement 2*). In contrast, the AR3A and AR3C CDRH2 contacts are reduced to hydrophobic residues in α1-helix (*Kong et al., 2013*), whereas the HEPC3 and HEPC74 CDRH2s contact the E2 α1-helix and the portion of the E2 front layer between the α1-helix and variable region 2 (residues 446–448) (*Flyak et al., 2018*).

A feature of *VH1-69*-derived antibodies is the presence of two hydrophobic residues at the tip of the CDRH2 loop that facilitate interactions with hydrophobic epitopes. The CDRH2s of AR3A and AR3C contain an Ile/Val-Pro-Met/Leu-Phe motif in which hydrophobic residues interact with the E2 front layer and CD81 binding loop (*Chen et al., 2019*). The CDRH2s of HEPC3 and HEPC74 are less hydrophobic and contain a Thr/Ser-Pro-Ile-Phe/Ser motif (*Chen et al., 2019*). In addition to hydrophobic interactions with the E2 front layer, the HEPC3 CDRH2 also makes a single hydrogen bond with E2 (*Flyak et al., 2018*). By contrast, AR3X is a not a typical *VH1-69* antibody in which hydrophobic residues in CDRH2 mediate the binding to hydrophobic residues in E2 (*Chen et al., 2019*). Instead, the AR3X CDRH2 forms eight hydrogen bonds with the E2 glycoprotein, four of which are mediated by AR3X residue Arg52g (AR3X-E2ecto: Pro52c-His445, Pro52e-Thr444, Arg52g-Ala440, Arg52g-Phe442, Arg52g-Tyr443, Arg52g-Pro612, Asn52n-Tyr443, Trp52i-Tyr613) (*Figure 4F*, *Figure 4—figure supplement 2*). Notably, these differences in binding interactions have functional implications, as these mAbs differ in potency of neutralization of individual HCV strains in the HCVpp panel. For example, the AR3X neutralization $IC_{50}$ for strain 1b21 is ~17 fold lower than the $IC_{50}$ of HEPC3 (1.2 vs. 20.5 µg/mL). In contrast, the AR3X neutralization $IC_{50}$ for strain 1a142 is ~9 fold higher than the $IC_{50}$ of HEPC3 (16.2 vs. 1.9 µg/mL) (*Figure 2*; *Flyak et al., 2018*).

A signature feature of the AR3A/AR3C and HEPC3/HEPC74 types of HCV bNAbs is the long CDRH3 that forms multiple main chain–main chain hydrogen bonds with E2 front layer residues (*Flyak et al., 2018*; *Tzarum et al., 2019*). Similar to other front layer-specific bNAbs with a CDRH3 disulfide motif, the first cysteine residue of the AR3X CDRH3 (Cys100a) hydrogen bonds with E2 residue Cys429 (*Figure 4G*, *Figure 4—figure supplement 2*; *Flyak et al., 2018*; *Kong et al., 2013*; *Tzarum et al., 2019*). Three additional hydrogen bonds (AR3X-E2ecto: Arg99-Asp431, Arg99-Asn430, Asn96-Asp431), as well as a salt bridge between CDRH3 (Arg100b) and a CD81 binding

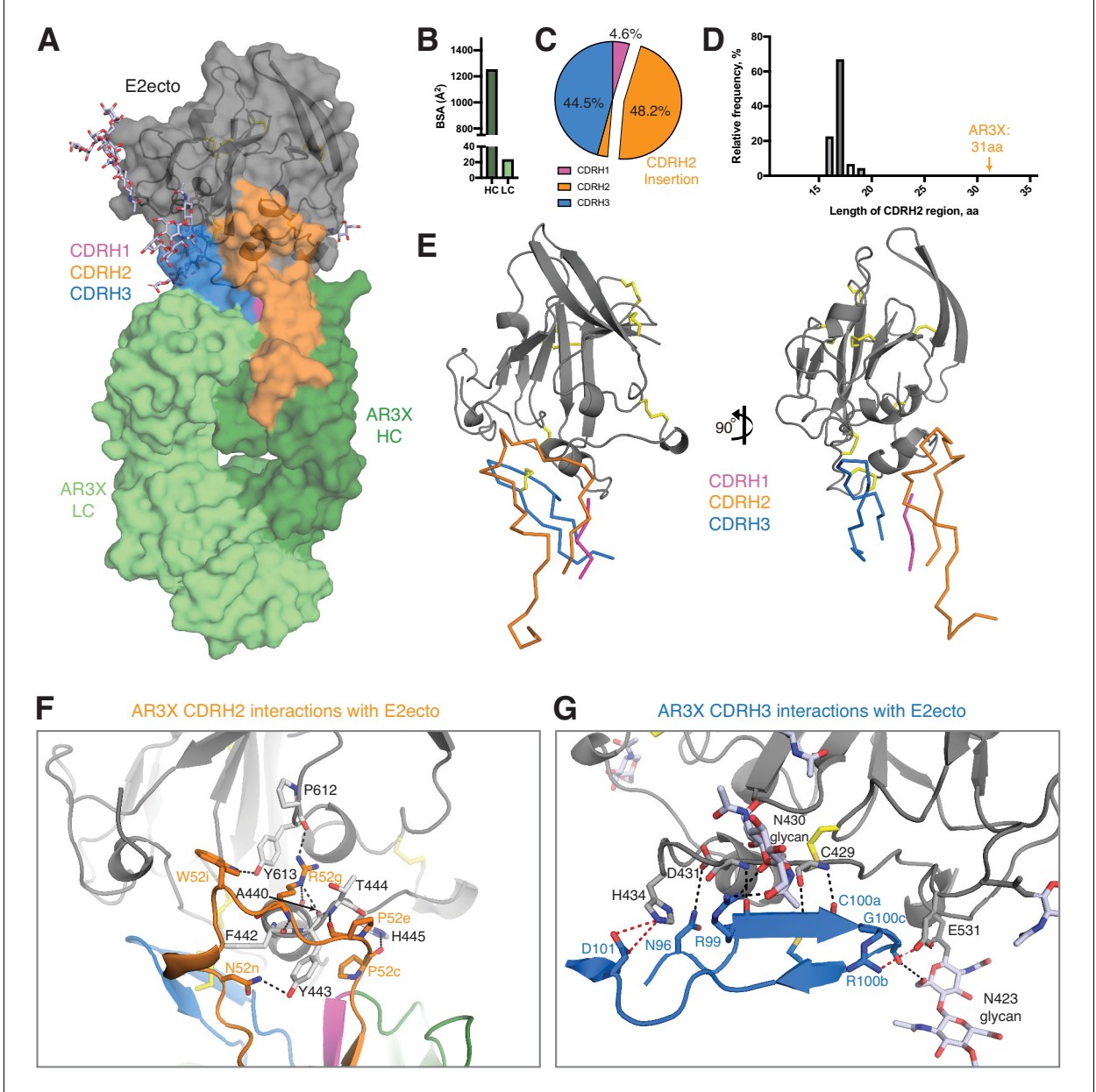

**Figure 4.** Details of the AR3X interactions with E2ecto. (a) Crystal structure of the AR3X-E2ecto complex. E2ecto is shown as a cartoon representation within a transparent surface with N-glycans highlighted as sticks and disulfide bonds shown as yellow sticks. The AR3X Fab is shown in a surface representation with highlighted CDRs. (b) Comparison of buried surface areas (BSAs) of E2ecto on the HC and LC of AR3X. (c) Percentage of BSA contributed from CDRH loops of the total BSA on the AR3X HC. The portion of CDRH2 within the CDRH2 insertion is separated from the main pie chart. (d) Length distribution of human CDRH2s. Human CDRH2 lengths were extracted from the online abYsis system (http://www.bioinf.org.uk/abysis/) using the Kabat numbering scheme *Kabat and National Institutes of Health (U.S.). Office of the Director, 1991*). (e) Interactions of AR3X heavy chain CDRs with E2ecto. CDRs are purple (CDRH1), orange (CDRH2), and blue (CDRH3) tubes. Disulfide bonds are shown as yellow sticks. (f) CDRH2 interactions with E2ecto. Interacting residues are shown as sticks. AR3X CDRH1 – purple, AR3X CDRH2 – orange, and AR3X CDRH3 – blue. Disulfide bonds are shown as yellow sticks. Potential H-bonds are shown as black dashed lines, and residues at the interface are indicated. (g) CDRH3 interactions with E2ecto. Interacting residues shown as sticks. For clarity, only the CDRH3 of AR3X is shown. Disulfide bonds are shown as yellow sticks and E2 glycans are shown as sticks with light blue, red, and dark blue colors for carbon, oxygen, and nitrogen atoms, respectively. Potential H-bonds and salt bridges are shown as black or red dashed lines, respectively. Residues at the interface are indicated.

The online version of this article includes the following figure supplement(s) for figure 4:

**Figure supplement 1.** Data collection and refinement statistics for AR3X-E2ecto1b09 complex.

**Figure supplement 2.** Interface residues between AR3X and E2ecto.

loop residue (Glu531), further stabilize the interaction of AR3X with E2. The AR3X-E2ecto crystal structure also shows contacts between the AR3X CDRH3 and N-glycans attached to E2 residues Asn423 and Asn430 (*Figure 4G*).

## Discussion

We and others previously described HCV bNAbs that utilize the *VH1-69* gene segment and a germ-line-encoded disulfide motif in CDRH3 to recognize the conserved epitope in E2 front layer (*Flyak et al., 2018*; *Keck et al., 2019*; *Kong et al., 2013*; *Tzarum et al., 2019*). Here we structurally characterized a front layer-specific HCV bNAb that is encoded by the *VH1-69* gene that includes an ultralong insertion in CDRH2 as well as the disulfide motif in CDRH3. We found that AR3X, isolated from the same chronically-infected patient as AR3A or AR3C (*Law et al., 2008*), surprisingly exhibits

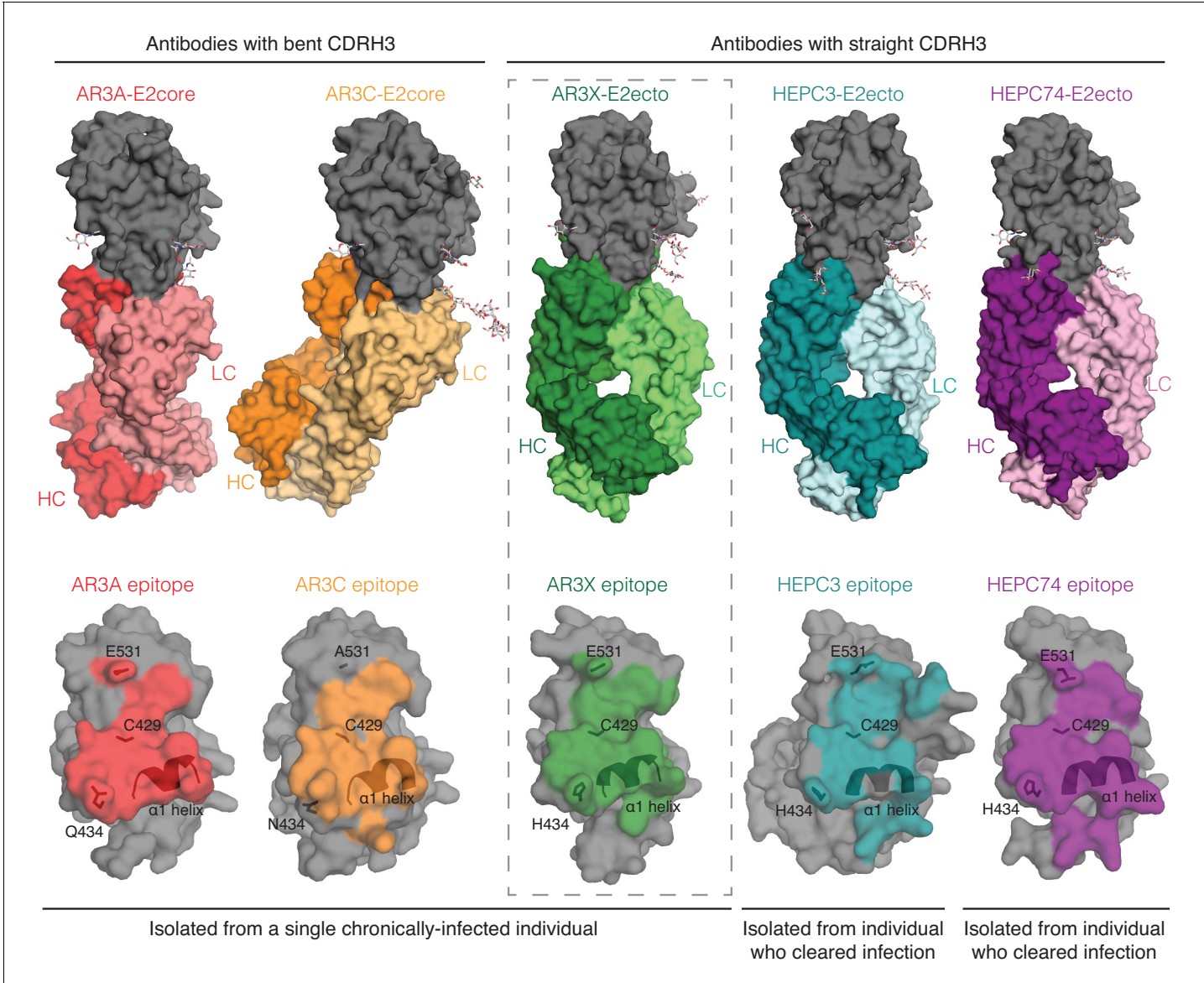

**Figure 5.** A structural plasticity of *VH1-69*-derived bNAbs with the CDRH3 disulfide motif. (Top) Surface representations of AR3X-E2 and other bNAb-E2 structures. E2, gray; AR3A-HC, red; AR3A-LC, light red; AR3C-HC, orange; AR3C-LC, yellow; AR3X-HC, green; AR3X-LC, light green; HEPC3-HC, blue; HEPC3-LC, light blue; HEPC74-HC, purple; HEPC74-LC, pink. (Bottom) Comparison of AR3A (red), AR3C (orange), AR3X (green), HEPC3 (blue), and HEPC74 (purple) epitopes. Epitopes on the E2 front layer (surface representation) were defined as residues in E2 containing an atom within 4 Å of the bound Fab.

the straight CDRH3 conformation found in the HEPC3 or HEPC74 bNAbs isolated from individuals who spontaneously cleared HCV infection (*Figure 3*). This indicates that a single individual can produce potent HCV-specific bNAbs using the common *VH1-69* and *D2-15* genes that bind to the conserved region of E2 in at least three different configurations (straight CDRH3 with CDRH2 insertion, straight CDRH3 without CDRH2 insertion, or bent CDRH3 without CDRH2 insertion), highlighting the intrinsic plasticity of the *VH1-69*–encoded CDRH1 and CDRH2 loops that accommodate different antibody approach angles (*Figure 5*). It's likely that the CDRH3s of these bNAbs dictate the preferential mode of engagement of bNAb germline precursors with the conserved epitope in the E2 front layer. Overall, these data demonstrate that B cells using VH1-69 and D2-15 genes can follow multiple pathways of affinity maturation to achieve broad neutralizing activity.

In the four bNAbs that were previously characterized structurally (*Flyak et al., 2018*; *Kong et al., 2013*; *Tzarum et al., 2019*), the first cysteine residue of the CDRH3 hydrogen bonds with E2 residue Cys429 (*Figure 4G*, *Figure 4—figure supplement 2*). We hypothesize that after the initial recognition of the front layer by CDRH3, the *VH1-69*-encoded CDRH1 and CDRH2 further stabilize the interaction while subsequent somatic mutations increase the bNAb affinity and breadth. Other antibodies that utilize a CDRH3 stabilized by a disulfide bond have been also described in the literature (*Sui et al., 2009*; *Thomson et al., 2008*; *Ying et al., 2015*). For example, M336, a potent human antibody that neutralizes severe acute respiratory syndrome coronavirus (*Ying et al., 2014*), is encoded by the *VH1-69* gene segment and includes a germline-encoded disulfide bond in its CDRH3 (*Ying et al., 2015*).

Nucleotide insertions and deletions play an important role in diversification of the antibody repertoire (*de Wildt et al., 1999*; *Reason and Zhou, 2006*; *Wilson et al., 1998*). Insertions are produced by sequence duplications; while the average size of insertion varies from 3 to 33 nucleotides, the majority of antibodies contain short insertions (*Kanyavuz et al., 2019*; *Wilson et al., 1998*). AR3X with its 42-nucleotide insertion in CDRH2 represents an interesting case of an antibody that utilizes an ultralong CDRH2 to bind its epitope (*Figure 4*). The insertion was required for recognition of E2 glycoproteins across multiple HCV strains, as evidenced by the poor binding activity of AR3X variants lacking the CDRH2 insertion (*Figure 2*). While several neutralizing antibodies with insertions have been described (*Kepler et al., 2014*; *Krause et al., 2011*), AR3X is unique for its exceptionally long CDRH2 insertion, which makes extensive contacts with E2, but does not change the preconfigured mode of AR3X interaction with E2 based on its straight CDRH3 containing a disulfide motif. Thus the conserved epitope in the HCV E2 front layer, which is recognized by multiple human bNAbs containing a disulfide motif in their CDRH3s (*Figure 5*), remains a promising target for lineage-based immunogen design.

## Materials and methods

**Key resources table**

| Reagent type (species) or resource | Designation | Source or reference | Identifiers | Additional information |
|---|---|---|---|---|
| Cell line (*Homo-sapiens*) | HEK293-6E | National Research Council of Canada | 11565 | |
| Cell line (*Homo-sapiens*) | Expi293F | Thermo Fisher Scientific | A14527 | |
| Cell line (*Homo-sapiens*) | Hep3B2.1–7 | ATCC | HB-8064 | |
| Antibody | Anti-Human IgG-HRP (Goat polyclonal) | SouthernBiotech | 2040–05 | 1:4000 dilution |
| Recombinant DNA reagent | pTT5 mammalian expression vector (used to express IgGs and Fabs) | National Research Council of Canada | N/A | |
| Commercial assay or kit | 1-Step Ultra TMB-ELISA Substrate Solution | Thermo Fisher Scientific | 34028 | |

*Continued on next page*

*Continued*

| Reagent type (species) or resource | Designation | Source or reference | Identifiers | Additional information |
|---|---|---|---|---|
| Commercial assay or kit | PEGRx HT | Hampton Research | HR2-086 | |
| Commercial assay or kit | PEG/Ion HT | Hampton Research | HR2-139 | |
| Commercial assay or kit | JCSG-plus HT-96 | Molecular Dimensions | MD1-40 | |
| Chemical compound, drug | Kifunensine | Sigma | K1140 | |
| Software, algorithm | Pymol | Schrödinger, LLC | RRID:SCR_000305 | |
| Software, algorithm | Phenix | (*Adams et al., 2010*) | https://www.phenix-online.org | |
| Software, algorithm | Coot | (*Emsley and Cowtan, 2004*) | http://www2.mrc-lmb.cam.ac.uk/personal/pemsley/coot/ | |
| Software, algorithm | PDBePISA | (*Krissinel and Henrick, 2007*) | http://www.ebi.ac.uk/pdbe/pisa/ | |
| Software, algorithm | abYsis system | | http://www.bioinf.org.uk/abysis/ | |
| Other | Superdex 200 Increase 10/300 GL | GE Healthcare | 17517501 | |
| Other | HisTrap FF column | GE Healthcare | 17531901 | |
| Other | HiTrap Protein A HP column | GE Healthcare | 17040301 | |
| Other | HCV 1b09 strain E1E2 sequence | GenBank | KJ187984.1 | |

## Cell lines

HEK293-6E cells were obtained from National Research Council of Canada. Expi293F cells were obtained from Thermo Fisher Scientific. Hep3B cells were obtained from American Type Culture Collection (ATCC). Hep3B cells were tested for mycoplasma contamination. Neither cell line is among the list of commonly misidentified cell lines.

## IgG expression and purification

Genes encoding the $V_H$ and $V_L$ domains of the AR3X bNAb called antibody 'A' in Supplemental Table 1 in *Law et al. (2008)* were synthesized as gBlocks gene fragments (IDT) and cloned into pTT5-based expression vectors (NRC Biotechnology Research Institute). Reverted unmutated ancestor (rua) variants of AR3X and the location of the insertion were inferred with IMGT/V-QUEST using complete sequences of heavy and light chain variable domains. IgGs were produced in Expi293F cells (National Research Council of Canada) by co-transfecting appropriate heavy and light chain plasmids. HiTrap Protein A HP column (GE Healthcare) was used to isolate IgGs from filtered culture supernatants followed by purification by size exclusion chromatography (SEC) using a Superdex 200 Increase 10/300 GL column (GE Healthcare).

## Expression and purification of E2 constructs

For ELISA experiments, His-tagged E2ecto proteins (residues 384–643) were expressed by transiently transfecting Expi293F cells (National Research Council of Canada) and purified from clarified supernatants using a HisTrap FF column (GE Healthcare) followed by SEC on a Superdex 200 Increase 10/300 GL column (GE Healthcare) to separate monomeric E2ecto proteins from oligomeric species. For structural studies, the His-tag was removed from an expression vector encoding a strain 1b09 E2 ectodomain.

## Expression and purification of an E2-Fab complex

AR3X Fab-1b09 E2ecto complexes for structural studies were produced in HEK293-6E or Expi293F cells (National Research Council of Canada) in the presence of 5 µM kifunensine (Sigma) by co-transfecting expression vectors encoding His-tagged Fab and untagged E2ecto to allow isolation of stable Fab-E2 complexes (*Flyak et al., 2018*). AR3X-E2 complex was purified from supernatants using Ni-NTA chromatography on HisTrap HP column (GE Healthcare) followed by SEC on a Superdex 200 Increase 10/300 GL column (GE Healthcare).

## Crystallization, data collection and structure determinations

Commercially-available screens (Hampton Research and Molecular Dimensions) were used to screen initial crystallization conditions by vapor diffusion in sitting drops. AR2X-E2ecto crystals were grown using 0.2 µL of protein complex in TBS and 0.2 µL of mother liquor (0.25 M ammonium tartrate dibasic pH 7.0, 20% PEG 3,350) and cryoprotected in mother liquor supplemented with 20% (w/v) glycerol. X-ray diffraction data from cryopreserved crystals were collected at the Stanford Synchrotron Radiation Lightsource on beamline 12–2 using a PILATUS 6M detector. Images were processed and scaled using iMosflm (*Battye et al., 2011*) and Aimless as implemented in the CCP4 software suite (*Evans and Murshudov, 2013*). The AR3X-E2 complex structure was solved by molecular replacement using the AR3C (PDB 4MWF) and 1b09 HCV E2ecto (PDB 6MEI) structures as search models. The models were refined and validated using Phenix.refine (*Adams et al., 2010*). Iterative manual model building and corrections were performed using Coot (*Emsley and Cowtan, 2004*). Glycans were initially interpreted and modeled using $F_o - F_c$ maps calculated with model phases contoured at 2σ, followed by $2F_o - F_c$ simulated annealing composite omit maps generated in Phenix in which modeled glycans were omitted to remove model bias (*Adams et al., 2010*). The quality of the final models was examined using MolProbity (*Chen et al., 2010*).

Models were superimposed and figures rendered using the PyMOL molecular visualization system (Version 1.7, Schrödinger, LLC). Buried surface areas (BSAs) were determined using the PDBePISA web-based interactive tool (*Krissinel and Henrick, 2007*). Potential hydrogen bonds were assigned using criteria of a distance of <4.0 Å and an A-D-H angle of >90°, and the maximum distance allowed for a van der Waals interaction was 4.0 Å. Rmsd calculations were done in PyMOL following pairwise Cα alignments without excluding outliers. AR3X residues were numbered according to the Kabat numbering scheme, and Kabat definitions of CDRs were used throughout the paper (*Kabat and National Institutes of Health (U.S.). Office of the Director, 1991*). Values to show the length distribution of CDRH2 in humans were extracted from the online abYsis system (http://www.bioinf.org.uk/abysis/) using the Kabat numbering scheme.

## ELISA binding analyses

Soluble forms of full-length E2 ectodomains were coated overnight onto 96-well plates (Corning) at 1 µg/mL. Plates were blocked with 1% goat serum and 1% powdered milk in TBST buffer (TBS with 0.05% Tween-20) for 1 hr. Purified IgGs were assayed in duplicate at 4-fold serial dilutions, starting at 10 µg/mL. IgGs-E2ecto complexes were detected using goat anti-human IgG horseradish peroxidase-conjugated secondary antibody (Southern Biotech, 1:4000 dilution) and 1-Step Ultra TMB-ELISA substrate (Thermo Fisher Scientific) and reading the optical density read at 450 nm after stopping the reaction with 1M HCl. A non-linear regression analysis was performed on the resulting curves using Prism version 5 (GraphPad) to calculate $EC_{50}$ values.

## HCVpp production and neutralization assays

HCVpp were produced by lipofectamine-mediated transfection of HCV E1E2 and pNL4-3.Luc.R-E-plasmids into HEK293T cells as described (*Hsu et al., 2003*; *Logvinoff et al., 2004*). A panel of 19 heterologous genotype 1 HCVpp has been described previously (*Bailey et al., 2015*; *Osburn et al., 2014*). Neutralization assays were performed as described (*Dowd et al., 2009*). Briefly, IgGs were serially diluted five-fold, starting at a concentration at 100 µg/ml and incubated with HCVpp for one hour prior to addition to Hep3B hepatoma cells. Luciferase activity was measured after three days and compared to that of HCVpp in media alone.

## Acknowledgements

We thank the Caltech Protein Expression Center (Dr. Jost Vielmetter, director) for help with protein expression and Dr. Anthony West for helpful discussions. Structural studies were assisted by the Caltech Molecular Observatory (Dr. Jens Kaiser, director). This research was supported by the National Institutes of Health grant R01 AI127469 (to JRB and PJB) (content is solely the responsibility of the authors and does not necessarily represent the official views of the NIH) and the Molecular Observatory at Caltech supported by the Gordon and Betty Moore Foundation. AIF was a Cancer Research Institute Irvington Fellow supported by the Cancer Research Institute. Use of the Stanford Synchrotron Radiation Lightsource, SLAC National Accelerator Laboratory, is supported by the U.S. Department of Energy, Office of Science, Office of Basic Energy Sciences under Contract No. DE-AC02-76SF00515. The SSRL Structural Molecular Biology Program is supported by the DOE Office of Biological and Environmental Research and by NIHGMS P41GM103393.

## Additional information

### Competing interests

Pamela J Bjorkman: Reviewing editor, *eLife*. Andrew I Flyak, Justin R Bailey: AIF and JRB are inventors of International Patent Application, Serial no. PCT/US2019/029315, pertaining to some of the antibodies presented in this article. The other authors declare that no competing interests exist.

### Funding

| Funder | Grant reference number | Author |
| --- | --- | --- |
| National Institutes of Health | R01 AI127469 | Justin R Bailey<br>Pamela J Bjorkman |
| Cancer Research Institute | Irvington Postdoctoral Fellowship | Andrew I Flyak |

The funders had no role in study design, data collection and interpretation, or the decision to submit the work for publication.

### Author contributions

Andrew I Flyak, Conceptualization, Data curation, Software, Formal analysis, Supervision, Validation, Investigation, Visualization, Methodology, Writing - original draft, Writing - review and editing; Stormy E Ruiz, Formal analysis, Investigation, Writing - review and editing; Jordan Salas, Semi Rho, Data curation, Formal analysis, Investigation, Writing - review and editing; Justin R Bailey, Supervision, Funding acquisition, Investigation, Writing - review and editing; Pamela J Bjorkman, Conceptualization, Supervision, Funding acquisition, Project administration, Writing - review and editing

### Author ORCIDs

Andrew I Flyak https://orcid.org/0000-0002-8722-479X
Stormy E Ruiz http://orcid.org/0000-0003-0892-9626
Pamela J Bjorkman https://orcid.org/0000-0002-2277-3990

### Decision letter and Author response

Decision letter https://doi.org/10.7554/eLife.53169.sa1
Author response https://doi.org/10.7554/eLife.53169.sa2

## Additional files

### Supplementary files

• Transparent reporting form

## Data availability

Diffraction data have been deposited in PDB under the accession code 6URH.

The following dataset was generated:

| Author(s) | Year | Dataset title | Dataset URL | Database and Identifier |
|---|---|---|---|---|
| Flyak AI, Bjorkman PJ | 2020 | Crystal structure of broadly neutralizing antibody AR3X in complex with Hepatitis C virus envelope glycoprotein E2 ectodomain | https://www.rcsb.org/structure/6URH | RCSB Protein Data Bank, 6URH |

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
