## [Decision Letter]

Thank you for submitting your article "An ultralong CDRH2 in HCV neutralizing antibody demonstrates structural plasticity of antibodies against E2 glycoprotein" for consideration by *eLife*. Your article has been reviewed by three peer reviewers, and the evaluation has been overseen by a Reviewing Editor and PÃ¤ivi Ojala as the Senior Editor. The following individuals involved in review of your submission have agreed to reveal their identity: Tongqing Zhou (Reviewer #1); Michael Houghton (Reviewer #2).

The reviewers have discussed the reviews with one another and the Reviewing Editor has drafted this decision to help you prepare a revised submission.

This work reported the structural and biochemical characterization of a neutralizing antibody AR3X isolated from a chronically HCV-infected individual. AR3X utilizes both its ultralong CDRH2 and a disulfide motif-containing straight CDRH3 to recognize the E2 front layer. Previous studies shown the structures of E2 complexes with front layer-specific bNAbs isolated from HCV-infected individuals, revealed a disulfide bond-containing CDRH3 that adopts straight (individuals who clear infection) or bent (individuals with chronic infection) conformation. The authors thus concluded that both the straight and bent CDRH3 classes of HCV bNAb can be elicited in a single individual, revealing a structural plasticity of VH1-69-derived bNAbs. This study is interesting and important for the readers interested in HCV vaccine development. There are some comments from the reviewers to be addressed, but no more experiments are needed.

Reviewer #1:

The manuscript by Andrew Flaky and colleagues titled "An ultralong CDRH2 in HCV neutralizing antibody demonstrates structural plasticity of antibodies against E2 glycoprotein" reported the structural and biochemical characterization of a neutralizing antibody AR3X isolated from a chronically HCV-infected individual, the crystal structure of antibody AR3X in complex with HCV E2 glycoprotein reveals some unusual features of antibody recognition of a conserved epitope on the E2 protein. Comparison with structures of other HCV E2-targeting antibodies showed distinct antibody binding poses. It has been observed that the disulfide bond-containing CDRH3 of HCV E2 front layer-targeting antibodies adopt a "straight" conformation while CDR H3 of antibodies isolated from chronically infected patients adopts a bent conformation. Structural analysis of AR3X, however, revealed its CDR H3 assumes a straight conformation, which indicates that both the straight and bent CDRH3 classes of HCV bNAb can be elicited in a single individual.

The paper is clearly written and presented an interesting structural phenomenon. However, it is not clear how these special new features relate to its biological function.

Specifically, I have some comments/questions:

1) Was AR3X broader in neutralization than others such as AR3A and AR3C?

2) Did the new structure features, such as long CDR H2 and straightened CDR H3, contribute to its breadth?

3) How does the epitope of AR3X look like on HCV E2? How does it compare to others'?

4) Is the footprint of AR3X bigger than the other ones? Does this make it harder for virus to escape?

5) The authors mentioned the different mode of recognition, a direct comparison of antibody mode of recognition by different antibodies on superposed on E2 will be much appreciated.

6) It will be helpful to readers if the authors show alignment of CDR H3 for AR3X and others. Is there a sequence feature that caused the "bent"?

7) Does the long CDR H2 affect the conformation of CDR H3 and make it straight?

Reviewer #2:

This very nice piece of work demonstrates that different HCV neutralising antibodies targeting the same conserved epitope isolated from the same patient bind in 3 different modes. The latter are due to the observed plasticity of the VH1-69 encoded CDRH1 and CDRH2 loops that facilitate different angles of antibody binding. This is important not just in demonstrating how neutralising antibodies mature in diverse ways in the same individual, but also in the context of using this information to design immunogens that could cross-neutralise a broader range of diverse HCVs. This approach could lead to an improved vaccine since the diverse antibodies have very different cross-neutralisation profiles.

Reviewer #3:

Flyak et al. analyzed the novel HCV E2 antibody, AR3X isolated from chronic carrier. AR3X utilizes both its ultralong CDRH2 and a disulfide motif-containing straight CDRH3 to recognize the E2 front layer. Previous studies shown the structures of E2 complexes with front layer-specific bNAbs isolated from HCV-infected individuals, revealed a disulfide bond-containing CDRH3 that adopts straight (individuals who clear infection) or bent (individuals with chronic infection) conformation. The authors thus concluded that both the straight and bent CDRH3 classes of HCV bNAb can be elicited in a single individual, revealing a structural plasticity of VH1-69-derived bNAbs. This study is interesting and important for the readers interested in HCV vaccine development. It is important to know the maturation process of B-cell which produce broadly neutralizing antibodies. The authors found the importance of both the CDRH2 insertion and the somatic mutations on AR3X binding and neutralization. Please clarify this unusual insertion is prerequisite for the somatic mutations or vice versa.

1) Figure 1D, indicated sequences of CDRH1, CDRH2, CDRH3 are different from their previous paper, Flyak et al., 2018. Please clarify the differences.

2) Figure 1C, there is no description about J3*02 in the text and legend.

3) Figure 2, it is interesting to know the effects of 17 somatic mutation and 14aa insertion for the biding and neutralization activities on other genotypes of E2 proteins.

---

## [Author Response]

Reviewer #1: The paper is clearly written and presented an interesting structural phenomenon. However, it is not clear how these special new features relate to its biological function.Specifically, I have some comments/questions:1) Was AR3X broader in neutralization than others such as AR3A and AR3C?

We thank the reviewer for the question. We added the following sentence to the main text: "The neutralization breadth of AR3X (89%) was slightly lower than the breath of AR3C bNAb (100%) (Flyak et al., 2018), which was isolated from the same HCV-infected individual (Law et al., 2008)."

2) Did the new structure features, such as long CDR H2 and straightened CDR H3, contribute to its breadth?

We edited the main text of manuscript to clarify the contribution of CDRH2 to the neutralization breadth of AR3X: "AR3X variants failed to neutralize HCV isolates, suggesting that both the CDRH2 insertion and somatic mutations are required for the *broad* neutralization activity of AR3X." Regarding the conformation of CDRH3 (straight or bent), we believe that the CDRH3 conformation does not directly translates into reduced or increased neutralization potency of an HCV-specific bNAb. Instead, the conformation of CDRH3 dictates the preferred mode of engagement of bNAb germline precursors with the conserved epitope in the E2 front layer, while subsequent somatic mutations and CDRH2 insertion further modulate the AR3X neutralization breadth.

3) How does the epitope of AR3X look like on HCV E2? How does it compare to others'?

The epitope of AR3X is similar to epitopes of other VH1-69 derived bNAbs with a CDRH3 disulfide motif. We added an additional figure to the manuscript (new Figure 5), which provides comparison of the AR3A, AR3C, AR3X, HEPC3, and HEPC74 epitopes.

4) Is the footprint of AR3X bigger than the other ones? Does this make it harder for virus to escape?

Overall, AR3X has a similar binding footprint to the footprints of HEPC3, HEPC74, AR3C, and AR3A, sharing multiple contact residues in the front layer and CD81 receptor-binding loop. We included this information in the main text (Results, fifth paragraph). In terms of the ability of HCV to escape from front layer-specific bNAbs, we believe that the amino acid variability within the epitope might impact the ability of the virus to escape neutralization by an antibody. For example, antibodies with smaller binding footprints that bind to the conserved region might display broader neutralizing activity than antibodies that make additional contacts through less conserved regions of E2 glycoprotein.

5) The authors mentioned the different mode of recognition, a direct comparison of antibody mode of recognition by different antibodies on superposed on E2 will be much appreciated.

We thank the reviewer for the suggestion. We now included the comparison of different bNAbE2 binding orientations in the new Figure 5, which also displays AR3X and other bNAb epitopes.

6) It will be helpful to readers if the authors show alignment of CDR H3 for AR3X and others. Is there a sequence feature that caused the "bent"?

We now included the CDRH3 alignment for AR3X and other bNAbs in Figure 1E. Both the position of disulfide motif relative to the whole CDRH3 as well as the presence of specific amino acids before or after the disulfide motif might be responsible for the straight/bent conformation of CDRH3. Further mutagenesis and structural studies are needed to determine the sequence motif responsible for distinct CDRH3 conformations.

7) Does the long CDR H2 affect the conformation of CDR H3 and make it straight?

We now included the CDRH3 alignment for AR3X and other bNAbs in Figure 1E. Both the position of disulfide motif relative to the whole CDRH3 as well as the presence of specific amino acids before or after the disulfide motif might be responsible for the straight/bent conformation of CDRH3. Further mutagenesis and structural studies are needed to determine the sequence motif responsible for distinct CDRH3 conformations.

Reviewer #3: Flyak, et al. analyzed the novel HCV E2 antibody, AR3X isolated from chronic carrier. AR3X utilizes both its ultralong CDRH2 and a disulfide motif-containing straight CDRH3 to recognize the E2 front layer. Previous studies shown the structures of E2 complexes with front layer-specific bNAbs isolated from HCV-infected individuals, revealed a disulfide bond-containing CDRH3 that adopts straight (individuals who clear infection) or bent (individuals with chronic infection) conformation. The authors thus concluded that both the straight and bent CDRH3 classes of HCV bNAb can be elicited in a single individual, revealing a structural plasticity of VH1-69-derived bNAbs. This study is interesting and important for the readers interested in HCV vaccine development. It is important to know the maturation process of B-cell which produce broadly neutralizing antibodies. The authors found the importance of both the CDRH2 insertion and the somatic mutations on AR3X binding and neutralization. Please clarify this unusual insertion is prerequisite for the somatic mutations or vice versa.

We agree with the reviewer that it is an interesting point. However, since the insertions are introduced during somatic hypermutation, it is challenging to determine whether the ultra-long insertion in CDRH2 of AR3X happens before, after, or at the same time as other somatic mutations are introduced. AR3X, along with AR3A and AR3C, was isolated from an individual who was chronically infected with HCV. By sequencing the antibody repertoire of this individual, one can potentially determine the architecture of AR3X lineage from which we could then infer events leading to the development of mature AR3X. Unfortunately, we do not have access to human samples that would enable such experiments, and such studies are outside the scope of this manuscript.

1) Figure 1D, indicated sequences of CDRH1, CDRH2, CDRH3 are different from their previous paper, Flyak et al., 2018. Please clarify the differences.

The observed differences are due to the different systems used to define CDR loops between two manuscripts. In our previous paper (Flyak et al., 2018), CDR loops were defined based on IMGT nomenclature. In the current manuscript, Kabat definitions of CDRs were used throughout the paper to allow a direct comparison of the AR3X structure with the recently-published structure of AR3A (Tzarum et al., 2019). The use of the Kabat numbering scheme is mentioned in the Materials and methods and in the legend to Figure 1.

2) Figure 1C, there is no description about J3*02 in the text and legend.

We added the description of J3*02 to the main text (Results, first paragraph).

3) Figure 2, it is interesting to know the effects of 17 somatic mutation and 14aa insertion for the biding and neutralization activities on other genotypes of E2 proteins.

This is an excellent suggestion. We have now evaluated the binding of AR3X and AR3X variants to a panel of E2 proteins representing six HCV genotypes (see updated Figure 2A and Figure 2—figure supplement 1). We also updated the main text to discuss the new results (Results, second paragraph). In summary, only mature AR3X recognized E2 glycoproteins from genotypes 2-6. AR3X DINS, AR3Xrua + INS, and AR3Xrua did not bind to E2 proteins from genotypes 2-6. These results further support our hypothesis that immunogens based on the genotype 1 1a157 ectodomain sequence could be used to stimulate the development of potent front layer-specific bNAbs.